# Current Strategies to Enhance Adipose Stem Cell Function: An Update

**DOI:** 10.3390/ijms20153827

**Published:** 2019-08-05

**Authors:** Yoojin Seo, Tae-Hoon Shin, Hyung-Sik Kim

**Affiliations:** 1Dental and Life Science Institute, Pusan National University, Yangsan 50612, Korea; 2Department of Life Science in Dentistry, School of Dentistry, Pusan National University, Yangsan 50612, Korea; 3Translational Stem Cell Biology Branch, National Heart, Lung, and Blood Institute, National Institutes of Health, Bethesda, MD 20892, USA

**Keywords:** adipose stem cells, mesenchymal stem cells, immunomodulation, cell therapy, function enhancement

## Abstract

Mesenchymal stem cells (MSCs) emerged as a promising therapeutic tool targeting a variety of inflammatory disorders due to their multiple remarkable properties, such as superior immunomodulatory function and tissue-regenerative capacity. Although bone marrow (BM) is a dominant source for adult MSCs, increasing evidence suggests that adipose tissue-derived stem cells (ASCs), which can be easily obtained at a relatively high yield, have potent therapeutic advantages comparable with BM-MSCs. Despite its outstanding benefits in pre-clinical settings, the practical efficacy of ASCs remains controversial since clinical trials with ASC application often resulted in unsatisfactory outcomes. To overcome this challenge, scientists established several strategies to generate highly functional ASCs beyond the naïve cells, including (1) pre-conditioning of ASCs with various stimulants such as inflammatory agents, (2) genetic manipulation of ASCs and (3) modification of culture conditions with three-dimensional (3D) aggregate formation and hypoxic culture. Also, exosomes and other extracellular vesicles secreted from ASCs can be applied directly to recapitulate the beneficial performance of ASCs. This review summarizes the current strategies to improve the therapeutic features of ASCs for successful clinical implementation.

## 1. Introduction

Mesenchymal stem cell (MSC) research in the field of cell-based therapy made great strides over recent decades due to their distinctive biological properties. MSCs, also named multipotent stromal cells, are adult fibroblast-like adhesive cells with self-renewal, multi-lineage differentiation potential [1]. Their outstanding proliferation and differentiation capabilities rendered MSCs widely applicable to regenerative medicine for the purpose of repairing and replacing damaged tissues [2]. Moreover, since their low immunogenicity and unique immunomodulatory properties were revealed [3,4], MSCs have been actively investigated as a cellular medicine for the treatment of a wide range of immune-related diseases, including graft-versus-host disease (GvHD), multiple sclerosis (MS), type I diabetes, rheumatoid arthritis (RA), periodontitis [5,6,7], inflammatory bowel disease [8,9,10,11,12], and even allergic dermatitis and asthma [13,14,15].

Although most progress on MSC-based cell therapy was achieved using MSCs isolated from bone marrow (BM), which is the first and prevailing source of MSCs, there is an increasing importance of non-BM tissues as an alternative MSC source. It was reported that MSCs can be obtained from a great number of tissue sources, including almost all adult tissues (e.g., fat, gingiva, skin, tonsil, and salivary gland), perinatal tissues (e.g., placenta, umbilical cord, and cord blood), and even pluripotent stem cells. In particular, among the various tissue sources of MSCs, adipose tissue emerged as an affluent and stable source of MSCs in terms of large quantity, high yield, ease of access, less invasive procedures and no ethical issues [16]. Taking advantage of these benefits and excellent MSC properties, adipose-derived stem cells (ASCs) or the fractions containing ASCs were extensively applied to treat various degenerative or inflammatory diseases with promising results [12,17,18,19,20,21].

Despite favorable pre-clinical data, however, ASCs showed somewhat limited efficacy and they fell short of expectations in advanced clinical trials (highlighted in Reference [22]). To exert the desired therapeutic potency, ASCs must meet several sets of conditions in the inhospitable host microenvironment. Once administered, ASCs should (1) survive long enough to (2) effectively migrate to the site of injury, and (3) fully exercise their desired functions or biological activities (Figure 1). It was reported that most of the injected MSCs disappeared after 24 h and only fewer than 1% of the cells survived for more than a week after systemic administration [23,24]. Thus, weak functionality/biological activity or the poor survival/migration capability of the injected ASCs may be the major cause of unsatisfactory clinical results. In addition, MSCs are highly heterogeneous cell populations with diverse phenotypic and functional characteristics depending on donors, tissues, and even subpopulations within the same origin [25,26]. Currently, ASCs are globally identified based on the minimal criteria provided by the International Federation for Adipose Therapeutics and Science (IFATS) and International Society for Cellular Therapy (ISCT) in 2013: (1) plastic adherence cells with self-renewal capability; (2) expression of cluster of differentiation 90 (CD90), CD73, CD105, and CD36, and lack of CD11b, CD45, CD31, and CD106; (3) differentiation potential into osteogenic, adipogenic, and chondrogenic lineages [27]. However, these criteria do not reflect the heterogeneity of ASCs [28], and the phenotypic markers or culture methods that can purify a homogeneous subset with outstanding biological functions are not fully elucidated. Therefore, many studies are being conducted to identify specific subpopulations of ASCs from different sources and to standardize the isolation and culture protocols.

Given the fact that the dose or frequency of ASC therapy cannot be increased infinitely in terms of cost and safety issues, strategies to boost the potency-related ASC functions prior to administration can serve as an attractive alternative to overcome the limited efficacy of naïve ASCs. Many attempts have been conducted so far, which include pre-conditioning or licensing ASCs with specific molecules, modification of genes of interest, and variation of culture conditions such as three-dimensional (3D) aggregates. It is also possible to directly adapt extracellular vesicles (EVs) secreted from ASCs instead of the ASCs themselves. Each strategy is not always individual, and integrated research is also underway to find optimal combinations across multiple methodologies. Increasing evidence supports the fact that these enhancement strategies provide higher efficiency compared to naïve cells. In this review, we collect and overview the strategies proposed to date for enhancing ASC function and therapeutic potency, dividing them into four categories: (1) pre-conditioning, (2) genetic manipulation, (3) culture condition modification, and (4) EV application.

## 2. Adipose Stem Cells

### 2.1. Characteristics of ASCs

ASCs refer to multipotent stem/stromal cells present within adipose tissues. Various terminologies were proposed to describe these cells but, currently, all types of these cells are uniformly called ASCs or ADSCs according to IFATS’s recommendation.

Adipose tissues are loose connective tissues which are largely divided into subcutaneous and visceral compartments. Each adipose tissue consists of several depots according to the anatomical location where it exists. In addition, there are two types adipose tissues (white and brown), depending on the main parenchymal cells of the tissues, called adipocytes [29]. The characteristics of ASCs vary depending on the type and anatomical region of adipose tissue, and subcutaneous white adipose tissues are generally regarded as the main source of ASC isolation due to their excellent yield of more than 2% in the stromal vascular fraction (SVF) [30,31]. Compared with the relatively low yield in BM-MSCs, ranging only from 0.001% to 0.1% in the mononuclear fraction [32], adipose tissue can provide up to 500-fold more MSCs than from an equivalent amount of BM aspirates [33], supporting the notion that adipose tissue can be the most abundant and efficient source of ASCs.

Moreover, it was reported that the properties of ASCs can be affected by the acquisition methods of adipose tissues or ASC isolation procedures. With regard to harvesting adipose tissues, power-assisted liposuction methodologies showed higher proliferative potential and resistance to senescence in isolated ASCs than laser-assisted liposuction and surgical biopsy [34]. Although there is no standardized method for isolating ASCs, the protocol proposed by Zuk et al., which includes the step of collagenase enzymatic digestion to get SVF, and NH_4_Cl treatment to remove erythrocytes, is commonly used for ASC isolation [35]. Several approaches were attempted to obtain ASCs more suitably for clinical application, such as non-enzymatic methods and the use of hypotonic NaCl instead of NH_4_Cl; however, they are yet to demonstrate a distinct advantage [36,37].

Adipose tissue is an extremely heterogeneous tissue composed of various cell types, including ASCs, preadipocytes, adipocytes, fibroblasts, vascular smooth muscle cells, endothelial cells, and lymphocytes [38]. Multipotent stem cells are known to reside mainly in the perivascular region of white adipose tissues [39]. Indeed, CD146^+^ CD34^−^ pericytes and CD146^−^ CD34^+^ supra-adventitial adipose stromal cells were revealed as two distinct MSC subsets in human adipose tissue with similar characteristics to BM-MSCs [40]. Given the fact that ASCs primarily reside around vasculature in fat tissue, scientists sought to verify whether what we refer to as ASCs are inherent MSCs originated from adipose tissue or unknown cells from a mesodermal origin that migrate from peripheral blood to adipose tissue [41]. A series of comparative studies were conducted in MSCs from different origins, revealing evidence that ASCs are unique MSCs from adipose tissue based on a difference in the degree of differentiation potential and phenotypic marker expression [42,43]. Although a specific marker for ASCs is yet to be identified, the unique expression of CD10 and CD36 and lack of expression of CD106 can be criteria to distinguish them from BM-MSCs [27]. However, there is still controversy over the expression of specific marker candidates in ASCs; some studies reported positive CD34 expression on ASCs [44,45], whereas a few studies showed either its absence or disappearance following passaging or cryopreservation [46]. STRO-1, which is considered one of the stemness markers for MSCs, was found to be negative in ASCs, distinct from BM-MSCs [47]; however, more recently, there was a conflicting report that ASC subsets express STRO-1 [48]. One study suggested CD271 as an ASC-specific marker [49]. Further studies are warranted to identify the precise stem cell subpopulation within adipose tissue with regard to the phenotype and distinct biologic functions.

In addition, the conditions of the donor, such as age, weight, and diseases status, can negatively impact ASC characteristics [50]. ASCs obtained from old individuals exhibited decreased proliferation, reduced adipogenic differentiation potential, and weakened paracrine activity compared to those from younger donors [51,52]. However, contrary to BM-MSCs, ASCs showed the advantages of retaining their number in adipose tissue and relatively maintaining their differentiation capacity with aging [53,54]. Interestingly, ASCs from obesity and type II diabetes patients were revealed to be defective in functionalities, including multi-lineage potential and immunomodulation, contributing to obesity and obesity-related inflammation and insulin resistance [55,56]. Given that the characteristics of ASCs vary dramatically depending on the donor status, the application of pre-validated allogeneic stem cells rather than autologous cells can be justified, and ASCs may be the most promising source for an allogeneic MSC bank.

### 2.2. ASC Applications and Limitations

ASCs are widely applied in pre-clinical and clinical investigations with the therapeutic purpose of regenerative medicine and immunomodulation in both autologous and allogeneic settings. Either the heterogenous whole SVF, which is yielded via mechanical and enzymatic digestion, or further isolated and expanded ASCs can be directly used in clinical studies. Several previous reviews already described the safety and benefits of ASCs or SVF in regenerative medicine [57,58]. In terms of functional enhancement, there is a need to distinguish the results between these two populations and to pay more attention to the purified ASC population. ASCs were reported to efficiently enhance the regeneration of skin [59,60], cartilage [61,62], myocardial tissue [63], and bone [64]. Moreover, based on several comparative studies showing superior immunomodulatory activities to BM- or umbilical cord-derived MSCs [65,66,67], ASCs were also implicated in the treatment of various immune-related disorders. Indeed, ASCs remarkably attenuated excessive inflammation in MS [68], RA [69], Crohn’s fistula [19], and GvHD [21]. However, ASCs still exhibit only moderate benefits in clinical trials, and physicians and scientists are struggling to move forward to the advanced phases of clinical trials (phases III and IV). To translate ASCs into clinical practice successfully, it is required to establish a substantial method of functional improvement, together with an attempt to clarify the heterogeneous adipose tissue and its corresponding ASC population.

## 3. Function Enhancement Strategies

The therapeutic potency of ASCs presupposes the presence of as many functional cells as possible in the damaged tissue. To achieve this requirement, many scientists sought to optimize the manipulation and administration methodology of naïve ASCs, including source differences, delivery timing, administration route, and dosage. Nonetheless, these attempts did not yield the desired improvement. Therefore, there is a need for novel strategies to enhance the capability of survival, homing to the site of inflammation, and immunomodulatory properties. Until now, four different approaches with encouraging outcomes were proposed in ASC therapy: pre-conditioning with various bioactive molecules, genetic engineering of functional genes, modification of culture condition, and direct application of extracellular vesicles. In this section, we discuss and update each ASC enhancement strategy according to the categories.

### 3.1. Pre-Conditioning (Priming or Licensing)

Pre-conditioning of ASCs with specific biological and biochemical factors in vitro is one of the foremost approaches to improve the cell function and therapeutic potential. After administration, ASCs must engraft, survive for a sufficient period of time, migrate to the damaged site, and exert the desired medicinal activities, during which the host microenvironment may modulate the biological properties and functions of ASCs, as well as their therapeutic capability. In fact, similar to immune cells, ASCs were shown to recognize microenvironmental stimuli and remember them transiently, leading to a change of cell fate [70,71]. Therefore, this pre-conditioning approach, also called priming or licensing, was proposed as a potential technique to provoke short-term memory and prepare for a harsh host microenvironment. Pre-conditioning can alter various ASC functions, such as survival (apoptosis), differentiation, regeneration, and migration, by inducing phenotype changes, genetic modification, and signaling pathway activation; however, pre-conditioning with inflammatory cytokines or mediators to enhance immunomodulatory properties and therapeutic potential gained the most attention so far (summarized in Table 1). Now, we discuss several methods suggested to efficaciously prime ASCs prior to administration, focused on the therapeutic potency.

#### 3.1.1. Pre-Conditioning with Cytokines

It is generally accepted that the immunomodulatory effects of MSCs are not substantially exerted but activated only in response to surrounding inflammatory milieu [72]. Ideally, MSCs reaching the inflamed site should exert the maximal immunomodulatory effects in response to the local milieu, but many scientists did not observe such bright results, presumably due to the following reasons: (1) inadequate local cytokine concentration to instruct all MSCs, (2) lack of time to be licensed, and (3) MSC property changes in undesired directions. Considering that it is impossible to grasp flawless in vivo tracking and the exact degree of inflammation [73], priming MSCs with the influential inflammatory cytokines may be an alternative for overcoming this hurdle.

Licensing with inflammatory cytokines is considered the most common way to mimic the in vivo inflammatory milieu and boost the immune regulatory function of MSCs. Interferon gamma (IFN-γ), a leading proinflammatory cytokine against viral and bacterial infections, is a representative source for MSC priming for functional enhancement [74]. IFN-γ priming was demonstrated to upregulate the synthesis of several anti-inflammatory transcription factors, including indoleamine 2,3-dioxygenase (*IDO*), *cyclooxygenase 2 (COX-2)*, transforming growth factor beta (*TGF-β*), and hepatocyte growth factor (*HGF*) [72,75,76]. In addition, IFN-γ licensing can not only upregulate the expression of various immunomodulation-associated genes such as *HLA-DR*, *ICAM1*, *VCAM1*, *CCL8*, *CXCL9*, and *CXCL10* [77,78], but also trigger the expression of class I and II major histocompatibility complex (MHC), contributing to host defense against pathogens and peptide-specific CD4 T-cell activation [79]. Based on the discoveries from in vitro settings, IFN-γ-driven overexpression of IDO was revealed as the main effector pathway in dose-dependent T-cell inhibition of ASC [80]. In addition, IFN-γ pre-conditioned ASCs showed pronounced anti-proliferative effects on activated peripheral blood mononuclear cells (PBMCs) along with a significant upregulation of PD-L1 expression and COX-2-derived PGE2 secretion [81]. The therapeutic potential of ASCs exposed to IFN-γ was reported in airway inflammation. A single pre-treatment of IFN-γ-primed ASCs ameliorated experimental obliterative bronchiolitis via IDO-dependent suppression of T-cell infiltration and induction of regulatory T cells (Tregs) [82].

Tumor necrosis factor-alpha (TNF-α), mainly secreted by macrophages, is the other major inflammatory cytokine used to prime MSCs to enhance their therapeutic potential. Pre-conditioning of ASCs with TNF-α could promote bone generation by increasing proliferation, mobilization, and osteogenic differentiation, primarily through the activation of the extracellular-signal-regulated kinase (Erk) 1/2 and p38 mitogen-activated protein kinase (MAPK) signaling pathway [83]. ASCs primed with TNF-α increased the secretion of interleukin-6 (IL-6) and IL-8, resulting in promoting endothelial progenitor cell homing and stimulating angiogenesis in a murine ischemic hindlimb model [84]. TNF-α pre-conditioning exhibited anti-inflammatory effects through upregulation of immunomodulatory factors such as IDO, PGE2, and HGF; however, this was to a much lesser extent than IFN-γ [85]. In this respect, scientists applied simultaneous licensing with IFN-γ and TNF-α to compensate for the lower efficiency and to obtain additive effects. An in vitro BM-MSC study demonstrated that the combination pre-treatment made MSCs less effective at increasing cytokine production by CD3/CD28-activated PBMCs and more potent at suppressing T-cell proliferation [86]. Domenis et al. showed that this combination pre-conditioning of ASCs induced the production of several immunomodulatory mediators such as PGE2, IL-10, and chemokine CCL2 [87].

A series of studies were conducted on priming effects of various pro-inflammatory cytokines to elucidate the optimal combination of cytokines that can maximize the therapeutic effect. Murine ASCs primed with IFN-γ, TNF-α, and IL-17 attenuated hepatitis through inducible nitric oxide synthase (iNOS)-mediated higher T-cell suppression [88]. Human ASCs pre-conditioned with IFN-γ, TNF-α, and IL-6 showed enhanced immunosuppressive properties in vitro [89]. In contrast, pre-conditioning ASCs with a combination of IL-1β, IL-6, and IL-23 represented a similar suppressive effect on allogeneic T-cell proliferation to naïve ASCs [90]; however, these primed cells exhibited higher TGF-β and lower IL-4. Conversely, pre-treatment of anti-inflammatory cytokines can be beneficial in enhancing ASC therapeutic potential. Li et al. showed in a murine model of glioblastoma that pre-exposure of ASCs to TGF-β upregulated CXCR4 expression, leading to the enhanced homing ability to cancer tissue, significant improvements in anticancer effect, and prolonged survival rate [91].

Pre-conditioning with inflammatory cytokines is a promising way to improve therapeutic effectiveness for tissue injury and inflammatory disease. However, many questions still remain unanswered. Inflammatory cytokines may exert undesirable side effects. For instance, class I and II HLA molecules upregulated by cytokine priming can confer immunogenicity to the cells, which makes them vulnerable to host immune responses [92]. Moreover, further and intensive studies need to be conducted to identify optimal cytokine combinations and dose, verify the uniformity in efficacy, and prove the detailed mechanisms.

#### 3.1.2. Pre-Conditioning with TLR Agonists

Other inflammation inducers can be used as a priming source for MSCs to maximize their therapeutic capabilities. Toll-like receptors (TLRs) represent a subgroup of pattern recognition receptors (PRRs) that contribute to the defense mechanism via the innate immune system in response to pathogen-associated molecular patterns (PAMPs) or damage-associated molecular patterns (DAMPs). TLRs are transmembrane proteins expressed mainly on innate immune cells, such as macrophages, dendritic cells, neutrophils, and epithelial cells, which recognize endosomal or extracellular PAMPs and activate inflammatory responses to protect the host. In addition to the TLR, several classes of cytoplasmic PRRs, including Nod-like receptors (NLRs) and RIG-I-like receptors (RIGs), were identified and extensively investigated. Each PRR subtype has distinct ligand specificity and function; thus, the aberrant activation of, or mutation in, specific PRRs can be closely related to the pathogenesis of a particular inflammatory disorder. Several studies revealed that MSCs functionally express TLRs and NLRs, indicating that MSCs act as not only sensors against infections or injuries but responders to microenvironmental stimuli. To date, the expression of TLR1, 2, 3, 4, 5, 6, and 9 and the lack of TLR7, 8, and 10 were reported in human MSCs [93,94], and these TLR expression patterns may vary depending on the origin of MSCs. Although some contradictory reports exist (i.e., the absence of TLR9 on BM-MSCs [95]), ASCs share the same TLR pattern with BM-MSCs; however, Wharton’s jelly-derived MSCs (WJ-MSCs) showed a limited expression of TLR3 and negativity for TLR4 protein expression [96]. Very low expression of TLR4 was also described on umbilical cord-derived MSCs (UC-MSCs) [97]. Considering that each TLR potentially reacts to its own ligands existing in the host microenvironment, resulting in the modulation of MSC properties by activating its downstream signaling pathway, pre-conditioning with a specific TLR agonist could be an efficient regulatory approach to strengthen the therapeutic potential of ASCs [93]. Indeed, many studies on the effect of TLR activation on MSC features, including proliferation, differentiation, and immunomodulatory properties, were widely conducted to explore effective enhancement methods.

Ligation of TLRs with a specific agonist enables them to serve as modulators of ASC multi-lineage differentiation capacity. Cho et al. [93] showed that activation of TLR2 (by peptidoglycan, PGN) and TLR4 (by lipopolysaccharide, LPS) significantly enhanced osteogenic differentiation in a dose-dependent manner, whereas triggering TLR9 (by CpG oligodeoxynucleotide, CpG-ODN) inhibited osteogenesis and ASC proliferation. Activation of TLR3 (by polyinosinic–polycytidylic acid, poly I:C) and TLR5 (by flagellin) did not cause any changes. Adipogenesis of ASCs was remarkably inhibited only in the presence of PGN. Similarly, pre-conditioning ASCs with poly I:C and LPS enhanced osteogenic differentiation without any effects on adipogenic differentiation and self-renewal [98]. The degree of altered differentiation potential may be different depending on the MSC source, even if the same TLR is stimulated [99], and the impact on other lineages such as chondrogenic differentiation remains to be clarified. Thus, more in-depth comparative investigations are needed to solve this question.

In terms of immunomodulatory function, recent studies reported that TLR4 activation via LPS turned MSCs into a pro-inflammatory phenotype, whereas TLR3 activation by poly I:C modified MSCs into a suppressive phenotype [71,100]. Pre-conditioning with low-dose LPS limited the immunosuppressive effects of ASCs by increasing the concentration of pro-inflammatory cytokines, such as IL-6 and TNF-α, and, at the same time, increased growth factors, including HGF and vascular endothelial growth factor (VEGF), enabled ASCs to improve liver function and regeneration in a mice hepatectomy model [101]. On the other hand, Mancheno-Corvo and colleagues showed that priming ASCs with poly I:C, but not IFN-γ, led to increased production of PGE2 [102]. However, the opposite result was also documented. Serejo et al. claimed that the activation of TLR3 signaling in ASCs, unlike BM-MSCs [103], had no effect on the immunosuppressive effect [104]. Therefore, it remains controversial whether licensing TLR3 enhances or reduces immunosuppression by MSCs. However, more encouraging results were documented when using the TLR ligands in combination with other cytokines, rather than their use as a single factor. The combination of LPS and IL-1β showed the most pronounced PGE2 production [105]. More studies are needed to understand the mechanisms of licensing in vivo and to induce appropriate activities in the same context as the previous cytokine pre-conditioning. Moreover, since there are relatively few studies on the expression and function of innate immune receptors in ASCs, research on this should be increased.

#### 3.1.3. Pre-Conditioning with Other Pharmacological/Bioactive Molecules

Bioactive molecules other than cytokines and agonists against immunoreceptors can also enhance ASC function by pre-conditioning in particular cases. Liu and colleagues reported that pre-treatment with curcumin, a potent antioxidant with anti-inflammatory properties extracted from the spice turmeric, improved the retention of ASCs transplanted and facilitated myocardial restoration by reducing myocardial apoptosis and enhancing neovascularization [106], mediated by the activation of PTEN/Akt signaling and an increased level of VEGF, respectively. In addition, some researchers described that pre-treatment with a certain combination of drugs, chemicals, or hormones could strengthen ASC function more efficiently than naïve ASC treatment. For instance, ASCs pre-conditioned with both LL-37, a host defense peptide belonging to the cathelicidin family, and bioactive lipid sphingosine-1-phosphate (S1P) improved self-renewal capability and enhanced anti-inflammatory and angiogenic potential via downregulation of pro-inflammatory genes (i.e., *IL-1β*, *IL-6*, *IL-12*, and *CCL2*) and upregulation of angiogenesis-related genes (i.e., *VEGFA*, *PDGF*, *HGF*, and *Ang-1*), respectively. The dual pre-conditioned ASCs effectively attenuated rat pulmonary artery hypertension [107]. Although this strategy using the pharmacological/bioactive molecules was not revealed much in comparison with that of inflammation-related substances, it is expected to provide important clues to identify the optimal “cocktail” combination to maximize ASC function.

### 3.2. Genetic Manipulation

Gene therapy is generally defined as the experimental technique that introduces exogenous DNA into a patient’s cells or directly in vivo to treat a variety of inherited and acquired genetic disorders. In recent years, the ease of MSC genetic manipulation was reported in vitro, and MSCs may serve as a suitable delivery vehicle for gene therapy [108,109]. Indeed, genetically modified MSCs, particularly overexpressing suicide genes, were used in glioblastoma repression [110], and genetic engineering can also be applied to transduce functionally critical genes directly into the ASCs themselves to enhance the therapeutic potency. To date, the genes responsible for survival, migration, and immunomodulatory properties were mainly targeted in ASC therapy and, overall, genetically modified ASCs were found to be more effective than wild-type cells [111]. Here, we introduce several reports on enhancing the function and therapeutic effect of ASCs through a genetic engineering approach.

**Table 1 ijms-20-03827-t001:** Pre-conditioning strategies to strengthen adipose-derived stem cell (ASC) function.

Priming Regimen	In Vitro Effects	In Vivo Effects	Model/Condition	Reference
IFN-γ	IDO ↑ T-cell suppression ↑	-	-	[80]
IFN-γ	PD-L1 ↑ COX-2/PGE2 ↑ PBMC proliferation ↓	-	-	[81]
IFN-γ	IDO ↑	T-cell infiltration ↓ Treg induction ↑	Obliterative bronchiolitis model	[82]
TNF-α	Proliferation ↑ Mobilization ↑ Osteogenesis ↑	-	-	[83]
TNF-α	IL-6 secretion ↑ IL-8 secretion ↑	EP cell homing ↑ Angiogenesis ↑	Ischemic hind limb model	[84]
IFN-γ/ TNF-α	PGE2 ↑ IL-10 ↑ CCL2 ↑	-	-	[85]
IFN-γ/ TNF-α IL-17	iNOS ↑	T-cell suppression ↑	ConA-induced hepatitis model	[88]
IFN-γ/ TNF-α/ IL-6	IDO ↑ Proliferation ↓ Cell diameter ↑ PBMC proliferation ↓	-	-	[89]
IL-1β/ IL-6/ IL-23	No morphologic change CD45 expression ↑ Differentiation ↑ Allogeneic T-cell proliferation ↓ TGF-β ↑, IL-4 ↓	-	-	[90]
TGF-β	CXCR4 ↑	Cancer homing ↑ Tumor volume ↓ Prolonged survival time ↑	Glioblastoma	[91]
LPS	IL-6, TNF-α, HGF ↑ VEGF ↑	Liver regeneration ↑ Serum AST, ALT ↓	Partial hepatectomy model	[101]
PolyI:C	PGE2 ↑ IDO activity ↑ Restoring ASC inhibitory effect on pre-stimulated T cells	-	-	[102]

↑; upregulated or enhanced, ↓; downregulated or reduced, -; not applicable.

#### 3.2.1. Genetic Modification to Enhance Retention and Migration

Since transplanted cells are vulnerable to the harsh microenvironment faced, most cells are cleared or turned to be dysfunctional in a short period of time, thus hindering their migration to the target site and them exerting their function. Therefore, prolonging the retention and improving migration capability are particularly important to improving the therapeutic efficiency of ASCs. The integration of the genes responsible for anti-apoptosis, self-renewal, and homing can be a favorable tool to achieve this goal. In addition, several efforts were made to increase the expression of multi-lineage differentiation and immunomodulatory genes in ASCs.

Sox2 and Oct4 are transcription factors that contribute to the maintenance of pluripotency in embryonic stem cells, as well as reprogram somatic cells into induced pluripotent stem cells. The forced expression of these two stemness-related genes was reported to enhance proliferation and prevent cell senescence in ASCs. Han et al. showed that ASCs overexpressing Sox2 and Oct4 exhibited enhanced proliferation, as well as osteogenic and adipogenic differentiation potential, in vitro by assigning ASCs a more primitive status [112]. Moreover, other studies achieved the overexpression of both genes in a vector-independent way, via a combination of leukemia inhibitory factor (LIF) treatment and stem cell-specific miR-302 transfection, and found beneficial effects such as improved proliferation and reduced oxidant-induced cell death [113,114]. In general, ASCs transduced with Sox2 and Oct4 showed remarkable benefits in their proliferation capability; however, it is important to note that more attention needs to be paid to a few conflicting results regarding differentiation potential [115] and possible adverse effects such as tumor formation for clinical applications.

Upon hypoxic pre-treatment (the hypoxic pre-treatment strategy is discussed in Section 3.3.2.), increased expression of superoxide dismutase 2 (SOD2) confers resistance to hypoxic stress by eliminating excessive reactive oxygen species (ROS), suggesting that SOD2 can be a convincing target of the genetic modification approach to improve ASC survival. In fact, ASCs virally overexpressing SOD2 resulted in significant survival improvement compared to transfected wild-type ASCs in vitro and in an in vivo syngeneic mice transplantation model [116]. In an obsess diabetic mouse model, mice receiving SOD2-overexpressed ASCs exhibited beneficial outcomes in reducing adiposity and improving glucose tolerance through anti-oxidative and anti-inflammatory effects [117].

C–X–C chemokine receptor 4 (CXCR4) signaling in response to its specific ligand stromal-derived factor-1 (SDF-1) serves as one of the critical factors involved in cell migration [118]. SDF-1 is vigorously elevated under inflammatory or ischemic conditions, attracting preexisting or delivered stem cells expressing CXCR4 to regenerate the damaged tissue. Therefore, overexpression of CXCR4 can enhance migration and mobilization of ASCs through activation of the SDF-1/CXCR4 signaling pathway. ASCs following lentiviral transduction of CXCR4 showed strengthend proliferative and anti-apoptotic properties, as well as increased migration capability in vitro, possibly via the Erk pathway. [119]. The enhanced homing and engraftment by CXCR4 gene transduction was consistently demonstrated as a result of significant muscle tissue regeneration in a mouse limb ischemic model [120]. Similarly, overexpression of granulocyte chemotactic protein-2 (GCP-2/CXCL6) in ASCs demonstrated beneficial effects on an experimental myocardial infarction model. GCP-2 is another chemokine that contributes to tissue homeostasis, tumorigenesis, and angiogenesis, and ASCs with genetically overexpressed GCP-2 promoted the proliferation and migration capabilities that contribute to improving heart function [121].

#### 3.2.2. Genetic Engineering to Improve Immunomodulation

Unique immune-regulatory capacities are the most promising characteristics for the development and application of ASCs as a “medicinal drug” and, on the same line as pre-conditioning with pro-inflammatory cytokines, genetic modification approaches were undertaken to directly or indirectly enhance the major immunomodulatory mediators of ASCs. Based on the strengthened immunosuppressive effects on various immune cells in vitro, the enhanced therapeutic or prophylactic advantages of genetically modified ASCs were verified in several in vivo inflammatory disease models.

Several studies described that the incorporation of anti-inflammatory genes such as *IL-10* [122], *HGF* [123], *IDO* [124], and *Foxp3* [125] could improve the therapeutic potential of MSCs. In addition, since each disease has its own pathogenic mechanism, a variety of other genes can be targeted to elicit a specific beneficial effect on the particular disease. Payne et al. showed that ASCs engineered to overexpress *IL-4* exerted protective effects in mice with experimental autoimmune encephalomyelitis (EAE) [126]. Mechanistically, EAE as a model for MS is an antigen-driven autoimmune model characterized by abnormal differentiation and activation of CD4^+^ T cells biased toward the T helper 1 (Th1) and Th17 subsets. In contrast, IL-4 secreted from Th2 cells may act as an anti-inflammatory mediator that alleviates excessive Th1/Th17-induced inflammation in this model. Overexpression of the *IL-4* gene in ASCs resulted in a reduction of antigen-specific T-cell responses and a compensational shift from pro- to anti-inflammatory cytokine response when delivered at early stage. Other studies revealed that genetically modified ASCs attenuated autoimmune arthritis in mice. *CTLA4Ig*-modified ASCs ameliorated collagen-induced arthritis by reducing serum type II collagen (CII) autoantibodies and increasing the ratio of Treg (CD4^+^ CD25^+^ FoxP3^+^) cells versus Th17 cells [127]. Park et al. showed that the transduction of receptor for advanced glycation end products (sRAGE) led to higher expression of immunomodulatory factors in ASCs, including IL-10, TGF-β, and IDO, and enhanced migration capability. Moreover, arthritic IL-1Ra-knockout mice receiving sRAGE-overexpressed ASCs exhibited marked remission of inflammatory arthritis by downregulating Th17 cells and reciprocally upregulating Treg cells [128].

There were also several studies on the introduction of genes encoding relatively newly identified cytokines into ASCs and the subsequent improvement of immunomodulatory properties. Marinez-Gonzalez et al. modified ASCs overexpressing soluble IL-1 receptor-like-1 (sST2), a decoy receptor for IL-33, and observed more pronounced pulmonary inflammation suppression and intact alveolar architecture than in naïve ASCs in endotoxin-induced acute lung injury models [129]. This dramatic effect was attributed to synergy with the increased expression of immunomodulatory molecules, such as IDO, TNF-α-stimulated gene-6 (TSG-6), and CXCR4, in response to the local inflammatory environment, along with further inhibitory effects on IL-33, TLR4, and IL-1β production resulting from sST2 overproduction. Additionally, ASCs overexpressing IL-35 exhibited higher suppressive effects on CD4^+^ T-cell proliferation and IL-17 secretion compared with non-transfected MSCs in an in vitro coculture setting [130].

Although the beneficial function enhancement of genetically modified ASCs was demonstrated as summarized in Table 2, several limitations remain in their clinical application. The application of replication-defective viral vectors, such as lenti- and adenoviruses, is closely associated with safety concerns including potential tumorigenicity, toxicity, and immunogenicity [131]. Moreover, the long-term curative effects, particularly at an organ or systemic level, are yet to be fully addressed. Therefore, extensive systemic studies are necessary to accumulate substantial evidence for the clinical application of gene-manipulated ASCs, including (1) the development of advanced gene integration methods with safety and efficiency, and (2) the elucidation of systematic in vivo mechanisms of genetically modified ASCs.

#### 3.2.3. Genetic Manipulation to Induce Lineage Transdifferentiation

In addition to the three mesenchymal lineages, ASCs can also undergo transdifferentiation toward non-mesenchymal cell lineages, including myogenic, cardiac, endothelial, and neuronal cells, in response to the lineage-specific inducer [35,132], although there are somewhat controversial views on neural transdifferentiation (reviewed in Reference [133]). Lineage conversion can be achieved in vitro through the exposure of ASCs to extrinsic signaling molecules or via the modification of culture conditions, such as using specific biomaterials. Alternatively, genetic manipulation integrating key transcriptional factors into ASCs may be a better way to induce stable and effective lineage transdifferentiation [134]. Although there are still obstacles to be overcome such as the development of safe gene delivery methods and the selection of the most appropriate target gene, encouraging evidence was accumulated over MSCs from different sources. In this section, we summarize the approaches to genetic manipulation to induce lineage transdifferentiation via overexpression of transcriptional factors in ASCs.

In the case of cardiomyogenic lineage, the forced expression of *Tbx20*, a critical transcription factor that contributes to heart development and cardiomyocyte regeneration, efficiently induced expression of cardiomyogenic differentiation markers on ASCs at 14 days after transduction both at the RNA and protein level [135]. It might be necessary to evaluate the cardiomyogenic regenerative capacity of Tbx20-overexpressed ASCs in an ischemic heart disease animal model. To transdifferentiate ASCs toward a neural lineage, Tang et al. transduced the proneural transcription factor Neurogenin (Ngn2) into ASCs and evaluated the in vitro neural lineage differentiation capacity and in vivo functional recovery in rat spinal cord injury (SCI). Rats transplanted with Ngn2-transduced ASCs showed higher expression of the neuron-specific nuclear protein (NeuN) in the injured site and exhibited the most striking functional recovery of the hind limb [136]. To enhance myogenic differentiation, Goudenege and colleagues transduced the key myogenic gene *MyoD* into human multipotent adipose-derived stem cells (hMADS) and observed a marked myogenic differentiation capability in vitro. Importantly, local intramuscular injection of *MyoD*-overexpressed hMADS cells into the cryoinjured Rag2^−/−^ γC^−/−^ immunosuppressed mice significantly improved muscle repair with the increase in hMADS-derived muscle fiber [137]. Moreover, it was reported that ETS variant 2 (ETV2) overexpression in ASCs can generate functional and expandable ETV2-induced endothelial-like cells (EiECs), which is expected to be an alternative strategy to treat ischemic vascular disorders [138].

### 3.3. Modification of Cell Culture Conditions

Since ASCs are isolated from individuals, they must be expanded in vitro to obtain a sufficient number of cells prior to clinical translation. In general, stem cells reside in a specific functional unit called the “stem cell niche” in vivo, which consists of not only the stem cells themselves but also other supporting cells and the extracellular matrix (ECM) [139,140]. In addition, micro-environmental factors such as oxygen tension, metabolic balance, and concentration of signaling molecules must be fine-tuned to maintain the niche homeostasis. The stem cell niche is essential for the integrity of the stem cell population with a dynamic balance between quiescent and active status [141,142]. Thus, the establishment of niche-like culture conditions is important to produce clinical-grade ASC lines in the aspect of quality and quantity issues. Notably, several studies revealed that modifications in conventional culture protocol could potentiate the therapeutic efficacy of ASCs. Here, we introduce some of the widely used alternative culture methods: 3D spheroid formation and hypoxic treatment.

#### 3.3.1. 3D Spheroid Formation

In general, conventional cell culture is conducted under a two-dimensional (2D) system in which cells grow as monolayers; however, it is a highly artificial environment deficient in cell-to-cell or extracellular interactions, leading to a decrease in therapeutic performance, as well as stemness, of adult stem cells [143]. To overcome this issue, the 3D spheroid formation technique was applied. Spheroids are multicellular structures in which adherent cells are forced to aggregate with each other using a suspension culture system [144,145]. They consist of cells, the ECM, and paracrine factors with a spontaneous metabolic gradient of both nutrient and oxygen concentration, providing more superior in vivo mimicking models compared to monolayer expansion systems [144,146]. Until now, the spheroid method was widely applied to culture neural stem cells (neurosphere), embryonic stem cells (embryonic body), cancer cells (tumoroid), and other cells to study developmental and physiopathological cell-to-cell dynamics in vitro [144,147].

In terms of cell therapeutics, several comparison studies between 2D and 3D cultured MSCs demonstrated that the spheroid culture method could improve the therapeutic potential of MSCs compared to the conventional method [148,149]. In general, spheroid formation could not only potentiate the expansion but also the differentiative capacity of MSCs [148,150,151]. High-throughput assessment of the transcriptome and secretome of 2D or 3D cultured MSCs revealed the enhanced immunomodulatory and tissue regenerative potential of MSC spheroids compared with monolayer MSCs [148,149,152]. In addition, the altered expression pattern of adhesion molecules on MSC spheroids could increase the migration and homing efficiency of MSCs into the damaged site with an enhanced engraftment ratio after in vivo application [151,153,154].

Based on these findings, the therapeutic potential of 3D-cultured ASC spheroids was evaluated in various pre-clinical settings. From in vitro observations, the 3D culture system is known to (1) increase the expression of ECM molecules such as E-cadherin, fibronectin, and laminin, (2) enhance the proliferative capacity and survival, (3) alter the differentiation preference, and (4) increase the secretion level of therapeutic factors including anti-inflammatory and pro-angiogenic molecules in ASCs, as summarized in Table 3. Of note, several reports supported the view that spheroid formation could transform ASCs into more primitive stem cells. Cheng et al. revealed that the expression levels of mesenchymal lineage markers CD29, CD90, and CD105 were decreased while pluripotency-related markers Sox2, Oct4, Nanog, and SSEA-4 were increased upon ASC aggregation [151,153]. Zhang et al. reported similar observations showing that ASC spheroids readily expressed Sox2, Oct4, Nanog, and Rex1 proteins in contrast to monolayer ASCs [155]. The dedifferentiation-like process during spheroid formation enables ASCs to trans-differentiate into non-mesenchymal lineages such as ectodermal (e.g., Nestin^+^ neural cells)- and endodermal (e.g., Albumin^+^ hepatocyte)-derived cells, although other molecular and functional assessments of dedifferentiated cells must be conducted to confirm these findings [151,155].

Therapeutic advantages of multicellular ASCs were also proven in experimental in vivo disease models. The enhanced differentiation capacity into osteocytes and chondrocytes in ASC spheroids could be directly applied to bone and/or cartilage defects [156,161]. Both in acute and chronic skin wound models, ASC spheroids extensively stimulated angiogenesis and granulation tissue formation compared to singular ASCs, shortening the wound closure time [152,153,159]. Similarly, the enhanced pro-angiogenic capacity of 3D cultured ASCs could attenuate the ischemic damage in experimental models for an acute kidney injury and hindlimb ischemia [158,160,162]. Others reported that intra-pleurally introduced ASC spheroids stimulated the regeneration of the lung via suppressing tissue-detrimental matrix metalloprotease activity, as well as increasing the level of tissue-regenerative basic fibroblast growth factor, in elastase-mediated emphysema mice [157]. Collectively, ASC spheroids could maintain their enhanced therapeutic functions in vivo as observed in vitro.

Although there is a general consensus that 3D-cultured stem cells exhibit therapeutic advantages over monolayer cells, several technical points should be considered to maximize their potency. Until now, the hanging drop method and culture on chemically (e.g., chitosan-coated) and/or physically (e.g., concaved bottom) modified plates are the most widely used methods for spheroid generation on a laboratory scale [144]. Recently, an automated 3D-bioreactor system was introduced to spheroid generation, which will contribute to minimizing the labor-intensive, time-consuming 3D culture procedure [144,155,161]. The size and the total cell number of each spheroid, as well as culture duration, are other critical factors to consider; since the core of spheroids is often nutrient-deficient and hypoxic, excessively large spheroids tend to contain non-functional or dead cells [145]. In addition, a longer culture period leads to densely packed spheroid formation, which can influence the expression patterns of the ECM and other secretory factors [153]. Therefore, researchers and clinicians must determine the most effective culture protocol to obtain a large number of homogeneous spheroids with stable therapeutic potency.

#### 3.3.2. Hypoxic Treatment

In a general setting, cells are cultured in normoxia (~21%) in vitro; however, the optimal oxygen concentration varies across the tissues, and oxygen tension changes dynamically in vivo [163]. Of interest, fine-tuned regulation of oxygen tension within the stem cell niche seems to be essential to properly maintain stem cell functions [164,165,166]. The ASC niche is known to be hypoxic, usually around 5% O_2_, compared to highly perfused organs [167]. As an important component of the stem cell niche, a low level of O_2_ during culture would, thus, provide a favorable in vitro environment for ASCs via emulating physiologic conditions. In addition, a hypoxic microenvironment is one of the leading host conditions causing poor survival of transplanted ASCs. Low oxygenation in damaged tissue increases the production of mitochondrial ROS in ASCs, leading to unsatisfactory survival and therapeutic efficiency. Therefore, some scientists assumed that exposing ASCs to oxidative stress prior to delivery would adapt them to the hypoxic environment and improve therapeutic potential.

Several studies assessed the impact of hypoxia on ASC characteristics and functions. First of all, hypoxia could contribute to ASCs maintaining their stemness, as well as proliferative capacity, during culture. The expression levels of stemness markers including Oct4, Sox2, and Nanog were increased in ASCs cultured under hypoxic conditions [168,169]. In addition, the low concentration of O_2_ seems to promote the proliferation of ASCs with increased viability, leading to a higher yield of ASCs compared to normal O_2_ levels [168,170,171,172]. It was also demonstrated that hypoxic culture conditions could protect ASCs against the most common in vitro damaging factors, replicative senescence and cryopreservation [173,174]. Meanwhile, several studies evaluated the impact of O_2_ tension on the osteo-, adipo-, and chondrogenic differential potential of ASCs, although there were conflicting outcomes depending on the experimental settings such as O_2_ concentration and the duration of differentiation [168,175,176,177,178]. Other beneficial aspects of ASCs would be enhanced during hypoxia as well; Rhijn et al. showed that ASCs cultured under 1% O_2_ inhibited the proliferation of mitogen-stimulated CD4 and CD8 T lymphocytes efficiently compared with control ASCs, implying the increased immunomodulatory effect of hypoxia-pre-conditioned ASCs [179]. Hypoxia is known to improve the angiogenic capacity of ASCs via stimulating the secretion of pro-angiogenic factor vascular endothelial growth factor (VEGF) and other growth factors in vitro [180,181]. Finally, a low level of O_2_ during the culture significantly increased the mobilization of ASCs toward inflamed sites in a cell migration assay, as well as a Boyden transwell chamber assay [172,182].

The therapeutic potential of hypoxic-treated ASCs was also studied in vivo for further clinical applications. As summarized above, hypoxia enhances the viability and survival rate of ASCs in vitro; thus, these pre-conditioned cells would not only exhibit increased engraftment capacity but also promote the survival of neighboring cells after in vivo application [183,184]. In addition, ASCs cultured under low O_2_ could drive endogenous angiogenesis via providing an extra level of growth factors such as VEGF and hepatocyte growth factor (HGF), contributing to an amelioration of the ischemic lesions [183,184,185,186]. Several in vivo studies demonstrated that the enhanced paracrine effect of ASCs after hypoxic treatment would be beneficial to diabetes, liver failure, irradiation-mediated salivary gland damage, and even glioblastoma models [187,188,189]. These findings would emphasize the advantages of hypoxic pre-conditioning in ASC culture in terms of therapeutics.

To take advantage of hypoxic-treated ASCs in a more practical way, some issues should be resolved. In experimental settings, hypoxia culture conditions include a wide range of O_2_ tensions from 5% to <1%, and each concentration could stimulate different properties of ASCs. For instance, the differentiation potential of ASCs was altered significantly when ASCs were cultured under 2–5% O_2_ tension, while a lower O_2_ level seemed to have little impact [173]. Moreover, recent studies revealed that hypoxic stress is increased severely in the adipose tissue of obesity- and/or other metabolic disorder-affected individuals, emphasizing the need to distinguish physiological and pathological hypoxia [190]. Therefore, the optimal O_2_ concentration for maximizing each therapeutic stem cell’s capacity should be determined prior to clinical application. In addition, an advanced in vitro culture system for the maintenance of constant O_2_ levels is essential to produce quality-controlled hypoxic-treated ASCs, which would contribute to obtaining reliable and reproducible results.

#### 3.3.3. Other Culture Environmental Modifications

Since ASCs require several growth factors and supplements like other MSCs, researchers tried to modify basic culture media composition to establish chemically defined media [191]. Specifically, many attempts were conducted to replace fetal bovine serum (FBS), one of the most commonly used supplements, due to its economic and clinical limitations [192,193]. Xeno-free and serum-free products are preferred as FBS substitutes, although some of these new reagents seem to induce senescence and surface antigen changes in ASCs compared to FBS-cultured cells [191,194]. In this respect, standard evaluation criteria such as the expression pattern of surface markers, growth morphology, and growth rate should be established to assess the efficiency and safety of ASCs cultured with a new formula.

In addition, the physical and mechanical aspects of the culture environment also affect cell characteristics [195,196]. For example, the stiffness of culture matrices regulates the expression pattern of cell surface markers and determines the differentiation preference of ASCs in vitro. Banks et al. reported that mechanical cues combined with biochemical signals could manipulate the differential potential of ASCs [197]. In this report, they changed matrix stiffness with collagen–GAG biomaterial and immobilized various differentiation-related growth factors onto a culture plate. They found that the stiffest matrix (~5 MPa) could induce the osteogenic differentiation of ASCs regardless of the presence of growth factors, while biochemical stimulants modulated the differentiation direction between osteogenesis and adipogenesis under relatively lower stiffness (~2.85 MPa). Wen et al. also reported similar observations showing that higher and lower stiffness could induce osteogenic and adipogenic differentiation, respectively [198]. These findings could contribute to the optimization of cell differentiation protocols for the development of tissue engineering techniques.

### 3.4. Application of Extracellular Vesicles (EVs)

It was suggested that the communication between MSCs and neighboring cells is pivotal for demonstrating MSC-derived therapeutic actions. The intercellular interaction is achieved not only via direct cell-to-cell contact (juxtacrine signaling) but also via a paracrine mechanism. Recently, EVs were regarded as a key mediator of paracrine signaling within the extracellular space [199]. EVs are membrane-derived, lipid bilayer vesicular structures with various physiological functions. EVs are classified into several subtypes including exosomes and microvesicles according to their size, biogenesis pathway, and route of secretion. It was noted that EVs can reflect the biological status of parent cells since they contain bioactive molecules such as microRNAs, lipids, growth factors, chemokines, and cytokines. In this respect, it would not be surprising that purified EVs from the culture supernatant of MSCs could yield similar advantages to cells in many pathologic conditions [200,201,202]. Indeed, several reports showed that long-lasting therapeutic outcomes could still be observed despite the high clearance rate of MSCs after in vivo application [203,204], implying that MSC-derived secretory, soluble factors might play an essential role in cell therapy. More importantly, although the relevance of MSC application for incurable and intractable diseases was proven in a number of pre-clinical and clinical findings, conventional cell-based therapy has several limitations that need to be considered for practical application. For example, the quality of MSCs during the overall procedure from cell isolation, to expansion, storage, and transfer should be tightly controlled to maintain reliable therapeutic efficacy. Safety concerns regarding the ectopic differentiation of introduced MSCs, as well as unintended long-term inhibition of the recipient immune system, are other key issues to be monitored. Therefore, cell-free EVs attracted significant attention as an alternative therapeutic option in recent years.

In general, MSC-derived EVs (MSC-EVs) are isolated from MSC culture media via a serial ultracentrifuge procedure, then applied to the treatment of a wide range of abnormal pathologic conditions in which MSCs proved to be effective. Recent advances in “omics” technologies enabled researchers to define therapeutic candidates among MSC-secreted paracrine factors [205,206,207]. Since one of the most significant roles of EVs is to mediate the horizontal transfer of parent cell-derived signaling molecules to target cells [208], MSC-derived beneficial molecules such as TGF-β1 [209], IL-10 [210], PGE2 [211], NO [212], and IDO [210,213] could be delivered via EVs. Therefore, the therapeutic actions of MSC-EVs are largely dependent on their tissue regenerative and immunomodulatory capacity as MSCs (Table 4) [214,215].

Until now, the therapeutic potential of ASC-EVs was analyzed in a wide range of pathologic conditions. Notably, many studies focused on the evaluation the advantages of EVs for neurological disorders because the central nervous system is hard to target with conventional cell delivery methods due to the presence of a blood–brain barrier.

To mimic the specific neurodegenerative condition, researchers used in vitro or in vivo transgenic models. Farinazzo et al. briefly proved the neuroprotective effect of ASC-EVs for a neuroblastoma cell line and primary hippocampal neurons under oxidative stress [216]. Authors also showed that ASC-EVs could activate oligodendrocyte progenitors to regenerate the damaged myelin sheath after the treatment of a demyelinating agent during ex vivo cerebellar slice culture, implying that EVs can interact with both neuronal and non-neuronal cells. Bonafede et al. transfected motor neuronal cell line NSC-34 with mutant superoxide dismutase (SOD1) transgenes as observed in familial amyotrophic lateral sclerosis (fALS)-affected neurons, then assessed the neuroprotective impact of ASC-EVs in vitro [217]. In this model, exosomes could protect SOD1-mutated motor neurons against hydrogen peroxide-mediated necrosis in a dose-dependent manner as previously reported [216]. Katsuda et al. suggested the advantage of ASC-EVs for Alzheimer’s disease (AD) [218,231]. Since the most significant etiology of AD is extracellular plaque formation within the brain caused by the aberrant accumulation of amyloid-beta protein, resolution of cerebral plaque is the key therapeutic strategy for AD. Based on the fact that neprilysin, one of the intensively studied endopeptidases for amyloid-beta proteolysis, is highly expressed in ASCs compared to BM-MSCs, Katsuda and his colleagues investigated whether ASC-derived exosomes had an anti-AD effect. They confirmed that exosomes isolated from ASCs contain physiologically active neprilysin. In addition, both extracellular and intracellular amyloid-beta produced by the N2a neuroblastoma cell line was degraded upon ASC supernatant treatment, implying that ASC-EVs might transfer ASC-derived neprilysin to neural cells. Meanwhile, an inflammation-induced demyelinating disorder, multiple sclerosis (MS), could be another target disease for ASC-EV therapeutics. It was reported that intravenously injected ASC-EVs could improve motor function while preventing brain atrophy in the murine model of Theiler’s murine encephalomyelitis virus (TMEV) infection, in which progressive MS-like symptoms are reproduced [219]. Of interest, neuro-inflammatory signs such as glial cell accumulation and overexpression of pro-inflammatory cytokines were diminished in the EV-treated group compared to the vehicle-treated group. In addition, neural stem cell activity in the subventricular zone was found to be increased upon ASC-EV treatment. The therapeutic benefits of ASC-EVs were also evaluated in another widely accepted animal model for MS, experimental autoimmune encephalomyelitis (EAE) [220]. In this paper, EVs were administrated to EAE mice before or after the disease onset, then clinical and histopathological severity was scored. It was noted that ASC-EVs might successfully play preventive roles in the progress of behavioral defects and neuroinflammation; however, they failed to rescue already developed EAE symptoms. EVs seemed to reduce inflammation partially via preventing the CXCL12–VCAM-1 mediated T-lymphocyte activation in the affected spinal cord.

In addition, many studies demonstrated that ASC-EVs could exert broad immunomodulatory and tissue regenerative roles in other disease models. Indeed, ASC-EVs ameliorated the atopic dermatitis-like skin lesions of mice via reducing the infiltration of innate immune cells such as mast cells and eosinophils [222]. The proliferation and activation of T lymphocytes, one of the important components of adaptive immunity, was suppressed in the presence of ACS-EVs in vitro [221]. Others reported that ASC-EVs stimulated the wound repair process by regulating fibroblast migration and re-assembly of the extracellular matrix within the damaged lesion [223]. ASC-EVs could also provide protection against ischemic injury both in vitro and in a myocardial infarction mice model via stimulating Wnt/β-catenin signaling [224]. Moreover, the pro-angiogenic effect of ASC-EVs was demonstrated by Liang et al [225]. In this study, authors found that the messenger RNA (mRNA) level of angiogenesis markers including Ang-1 and Flk1 was increased in human umbilical vein endothelial cells treated with ASC-EVs. During both the in vitro tube formation assay and in vivo Matrigel plug assay, ASC-EVs stimulated the formation of a vascular-like tubular structure within the matrix through microRNA-125a transfer, an upstream negative regulator of anti-angiogenic Notch signaling, to endothelial cells.

Of note, modification and engineering techniques as described above could be applied to ASCs to produce superior EVs with higher therapeutic potential compared to naïve EVs. One paper demonstrated that MSCs primed with inflammatory cytokines IFN-γ and TNF-α could produce more immunosuppressive EVs compared to control MSCs, and basal proliferation levels of both innate (natural killer cells) and adaptive (T and B lymphocyte) immune cells declined more effectively in the presence of primed MSC-EVs than control MSC-EVs [227]. In another study, EVs isolated from hypoxic-cultured ASCs exerted greater protection on cardiotoxin-induced skeletal muscle damage via inducing the class-switch of macrophages from pro-inflammatory M1 type to immunomodulatory and regenerative M2 type [228]. Since most of the ASC-derived secretory molecules can be found in EVs, genetic modification aimed at the overexpression of therapeutic factors is also one of the preferred strategies for obtaining high-quality EVs. This strategy has several advantages, not only to avoid safety issues caused by genetic modifications but also to specify the mode of action for stem cell therapy. Yu et al. reported that EVs isolated from GATA-4-overexpressing MSCs prevented myocardial ischemic damage both in vitro and in vivo because they contained a higher level of anti-apoptotic microRNA miR-19a than control MSCs [226]. In other studies, microRNAs known to play beneficial roles in a specific pathologic condition were directly overexpressed in ASCs to produce microRNA-rich EVs. Lou et al. produced miR-122-rich ASC-EVs via ASC transfection with a miR-122 expression plasmid. Interestingly, the chemosensitivity of hepatocellular carcinoma (HCC) toward anticancer agent sorafenib was improved upon miR-122-rich ASC-EV administration compared to naïve ASC-EVs, implying their therapeutic potential as an anticancer agent [229]. Qu et al. conducted a similar study to target liver fibrosis with ASC-EVs rich in miR-181-5p, which is known to regulate autophagy. Upon treatment of powerful fibrosis-inducing cytokine TGF-β, hepatic stellate cells (HSCs) started to proliferate actively and produced a profound amount of ECM. It is noted that miR-181-5p-rich ASC-EVs suppressed HSC activation via directly targeting proliferative STAT3 signaling, leading to autophagy. The anti-fibrotic role of miR-181-5p-rich EVs was also proven in a CCl4-induced liver fibrosis model, where inflammatory factors and liver injury markers declined after modified EV injection. Therefore, it would be worthy to apply various strategies known to enhance the potency of ASCs to harvest therapeutically superior ASC-EVs. In addition, EV engineering techniques to improve targeting and migration efficiency, as well as further optimization of the EV handling process (from harvest and quantification to storage), should precede the practical translation of ASC-EVs.

## 4. Conclusions

Until now, the therapeutic benefits of ASCs were demonstrated in a variety of pathological conditions ranging from inflammatory to degenerative disorders, which increases the expectation for utilizing ASCs in translational medicine. Prior to clinical application, many efforts should be dedicated to establishing the criteria for the isolation, maintenance, characterization, and cryopreservation of ASCs to standardize cell quality. Also, there is a growing consensus on the need for innovative strategies to maximize the therapeutic potential of ASCs. As summarized in this review, the latest technologies such as genetic editing and advanced culture systems, combined with a comprehensive understanding of the therapeutic mechanisms of ASCs, would contribute to a significant improvement in the efficacy of ASC treatment. In addition, the development of standard platforms for evaluating the therapeutic characteristics of naïve and modified ASCs could provide a useful screening tool for the selection of clinically potent cells.

## Figures and Tables

**Figure 1 ijms-20-03827-f001:**
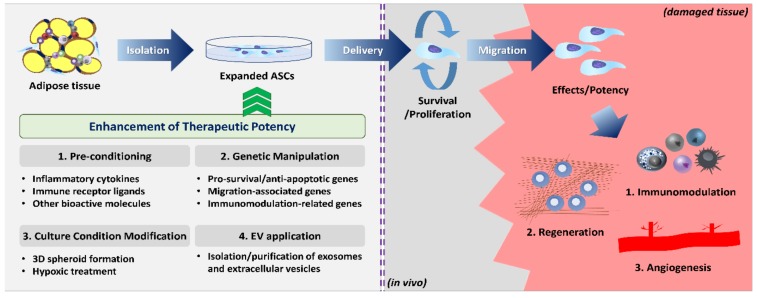
Overview of the action mechanisms and functional enhancement strategies of adipose-derived stem cell (ASC) therapy.

**Table 2 ijms-20-03827-t002:** Genetic manipulation approaches to enhance ASC function.

Target Gene	In Vitro Effects	In Vivo Results	Model/Condition	Reference
Sox2/Oct4	Proliferation ↑ Osteogenesis ↑ Adipogenesis ↑	-	-	[112]
SOD2	Survival ↑	Engraftment ↑	Hypoxia	[116]
SOD2	ROS ↓ PPARG, FABP-4, IL-6, TNFα expression ↓	Body weight ↓ Adipocyte inflammation ↓ Glucose tolerance ↑	Hyperglycemia (obese diabetic mouse model)	[117]
CXCR4	Proliferation ↑ Apoptosis ↓ Migration ↑	-	-	[119]
CXCR4	-	Long-term engraftment ↑ Muscular regeneration ↑	Diabetic mice with hindlimb ischemia	[120]
GCP-2/ CXCL6	VEGFA, HGF, IL-8 ↑ IGF-1, Akt-1 ↑ Proliferation ↑ Migration ↑ Endothelial differentiation ↑	Angiogenesis ↑ Infarct size ↓ Heart function ↑	Myocardial infarction model	[121]
IL-4	T-cell suppression ↑	MOG-specific T-cell priming ↓ EAE protective effect	Experimental autoimmune encephalomyelitis	[126]
CTLA4Ig	-	Treg/Th17 ratio ↑ CII autoantibodies ↓ CIA therapeutic effect	Collagen-induced arthritis	[127]
sRAGE	IL-1β, IL-6, VEGF ↓ IDO, IL-10, TGF-β, HGF ↑ Migration ↑	Treg/Th17 ratio ↑ Inflammatory arthritis ↓	Arthritic IL-1Ra-knockout mice	[128]
sST2	Immunomodulatory mediator expression ↑	Pulmonary inflammation ↓ Alveolar architecture ↑	Endotoxin-induced acute lung injury	[129]
IL-33	T-cell proliferation ↓ IL17 secretion ↓	-	-	[130]

↑; upregulated or enhanced, ↓; downregulated or reduced, -; not applicable.

**Table 3 ijms-20-03827-t003:** Summary of current three-dimensional (3D) spheroid formation methods and characteristics.

Culture Protocol				Phenotypic Changes	Applied Disease Model	Reference
**Method**	Duration	Size	Seeding Density			
Chitosan-coated culture plate	7 days	150 μm	2.5 × 10^4^ cells/cm^2^	ECM ↑ Pluripotent marker ↑ Osteogenesis ↑ Adipogenesis ↓	-	[151]
Chitosan-coated culture plate	7 days	-	2.5 × 10^4^ cells/cm^2^	Pluripotent marker ↑ Angiogenesis ↑	Skin wound-healing model	[153]
Silicon elastomer-based concave wells	5 days	~200 μm	1 × 10^5^, 3 × 10^5^, 6 × 10^5^/well	Osteogenesis ↑ Angiogenesis ↑	-	[156]
Polydimethylsiloxane-based concave wells	1 day	-	10^5^/well	Growth factor ↑ Pro-survival signal ↑	Elastase-induced emphysema model	[157]
Ultra-low attachment plates	3 days	~50 μm	-	ECM ↑ Angiogenesis ↑	Hindlimb Ischemia model	[158]
Ultra-low attachment plate	3 days	-	7.5 × 10^4^ cells/cm^2^	Angiogenesis ↑ endothelial markers ↑	Skin defect wound-healing model	[159]
Hanging drop	1 day	-	25,000 cells/drop	ECM ↑ Antioxdative effect ↑ Angiogenesis ↑	Acute kidney ischemia model	[160]
Hanging drop	1 day	~200 μm	25,000 cells/drop	ECM ↑ Angiogenesis ↑	Diabetic skin wound model	[152]
Spinner flask	3 days	>200 μm	10^6^ cells/mL	Chondrogenesis ↑	-	[161]
Microgravity bioreactor	5 days	123.4 ± 26.2 μm	10^6^ cells/mL	ECM ↑ Osteogenesis, Adipogenesis, Chondrigenesis ↑	Tetrachloride-induced acute liver failure	[155]

↑; upregulated or enhanced, ↓; downregulated or reduced, -; not applicable.

**Table 4 ijms-20-03827-t004:** Therapeutic impact of naïve and modified ASC-derived extracellular vesicles (EVs).

Priming Regimen	EV Source	In Vitro Effects	In Vivo Effects	Target Disease	Ref.
Filtration +UC	Naïve ASCs	Oxidative stress↓ in SH-SY5Y cells and primary murine hippocampal neurons	-	-	[216]
PureExo® Exosome isolation kit	Naïve ASCs	Oxidative stress↓ in NSC-34 cells	-	Familial ALS	[217]
Filtration + UC	Naïve ASCs	Amyloid-β levels↓in N2a cells		AD	[218]
UC	Naïve ASCs	-	Cell proliferation in the SVZ↑ Anti-inflammatory brain atrophy↓	MS	[219]
UC	Naïve ASCs	Proliferation↓in N9 cells activation and adhesion capacity↓in CD4+ T cells	Limiting immune cell infiltration Anti-inflammatory behavior improvement	MS	[220]
UC	Naïve ASCs	Proliferation↓in T cells inhibit IFN-γ production terminal differentiation↓in effector-memory T cells	-	-	[221]
UC	Naïve ASCs		Clinical score↓ Inflammatory dendritic epidermal cell/mast cell infiltration↓ Serum IgE level↓	Atopic dermatitis	[222]
Exoquick® Exosome isolation kit	Naïve ASCs	Migration, proliferation, collagen synthesis↑in fibroblast	Cutaneous wound healing↑ Collagen production↑	-	[223]
Filtration + UC	Naïve ASCs	Hypoxic damage and apoptosis↓in H9C2 cells	Infarct size↓Ischemic damage-related marker↓Myocardial apoptosis↓	Ischemic heart disease	[224]
Filtration + UC	miR-125a-OE ASCs	Pro-angiogenic gene expression↑in HUVEC vascular length and branch↑in tube formation assay	Vascular structure↑in Matrigel plug assay	-	[225]
Exoquick® Exosome isolation kit	GATA4 OE ASCs	Hypoxic apoptosis↓in cardiomyocyte	Infarct size↓ Cardiac fibrosis↓ Ventricle wall thickness↑	Regional myocardial ischemia	[226]
UC	IFNγ/TNFα primed ASCs	Proliferation↓in B cells and NK cells Immunosuppressive effect↑in MSCs	-	-	[227]
UC	Hypoxic(1%) cultured ASCs	Macrophagic M2 induction Macrophagic M1 inhibition	Vascular structure↑in Matrigel plug assay macrophage infiltration↓at the damaged site muscular regeneration↑	Skeletal muscle Injury	[228]
Exoquick® Exosome isolation kit	miR-122-OE ASCs	Chemosensitivity↑in HepG2 cells	Sensitization of HCC cells to Sorafenib tumor growth↓	HCC	[229]
Exoquick® Exosome isolation kit	miR-181-5p-OE ASCs	Proliferation↓in HST-T6 cells Autophagy↑in HST-T6 cells	Fibrosis↓ inflammation↓	Liver fibrosis	[230]

↑; upregulated or enhanced, ↓; downregulated or reduced, -; not applicable.

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
