# Peer review of "Current Strategies to Enhance Adipose Stem Cell Function: An Update"

_ijms, 2019, doi:10.3390/ijms20153827_

Round 1

Reviewer 1 Report

rows 75 - 76: "In addiction, ... differentiation capacity", add osteogenic, adipogenic and chondrogenic differentiation (Saler et al., hASC and DFAT, Multipotent Stem Cells for Regenerative Medicine: A Comparison of Their Potential Differentiation In Vitro, International Journal of Molecular Sciences, 2017, 18, 2699; doi:10.3390/ijms18122699)

row 76: "immunomodulatory properties" ... add references

rows 100 - 103: "The adipose ... of SVF" ... add more recent reference (Cinti, S. The adipose organ at a glance. Dis. Models Mech. 2012, 5, 588–594; Saler et al., hASC and DFAT, Multipotent Stem Cells for Regenerative Medicine: A Comparison of Their Potential Differentiation In Vitro, International Journal of Molecular Sciences, 2017, 18, 2699; doi:10.3390/ijms18122699)

rows 107 - 110: "ASC is positive for CD34" .... Literature data about the expression of CD34 in hASC are conflicting (Oedayrajsingh Varma, M.J.; Van Ham, S.M.; Knippenberg, M.; Helder, M.N.; Klein-Nulend, J.; Schoten, T.E.; Ritt, M.J.; Van Milligen, F.J. Adipose Tissue Derived Mesenchymal Stem Cell yield and growth characteristics are affected by the tissue-harvesting procedure. Cytotherapy 2006, 8, 166–177; Shah, F.S.; Li, J.; Zanata, F.; Curley, J.L.; Martin, E.C.; Wu, X.; Dietrich, M.; Devireddy, R.V.; Wade, J.W.; Gimble, J.M. The relative functionality of freshly isolated and cryopreserved Human Adipose-Derived Stromal/Stem Cells. Cells Tissue Organs 2016, 201, 436–444)

Author Response

Reviewer #1

rows 75 - 76: "In addiction, ... differentiation capacity", add osteogenic, adipogenic and chondrogenic differentiation (Saler et al., hASC and DFAT, Multipotent Stem Cells for Regenerative Medicine: A Comparison of Their Potential Differentiation In Vitro, International Journal of Molecular Sciences, 2017, 18, 2699; doi:10.3390/ijms18122699)

row 76: "immunomodulatory properties" ... add references

R: As the reviewer suggested, we specified three major mesenchymal lineages, osteogenic, adipogenic and chondrogenic, in the sentence and added relevant references.

rows 100 - 103: "The adipose ... of SVF" ... add more recent reference (Cinti, S. The adipose organ at a glance. Dis. Models Mech. 2012, 5, 588–594; Saler et al., hASC and DFAT, Multipotent Stem Cells for Regenerative Medicine: A Comparison of Their Potential Differentiation In Vitro, International Journal of Molecular Sciences, 2017, 18, 2699; doi:10.3390/ijms18122699)

R: We appreciate the correction of our referencing. In the revised manuscript, we cited more recent references regarding adipose tissue and cell population as suggested by the reviewer. Moreover, we added a brief description of the adipose tissue type, isolation technique, and characteristics of ASC in order to clarify the reader’s understanding.

rows 107 - 110: "ASC is positive for CD34" .... Literature data about the expression of CD34 in hASC are conflicting (Oedayrajsingh Varma, M.J.; Van Ham, S.M.; Knippenberg, M.; Helder, M.N.; Klein-Nulend, J.; Schoten, T.E.; Ritt, M.J.; Van Milligen, F.J. Adipose Tissue Derived Mesenchymal Stem Cell yield and growth characteristics are affected by the tissue-harvesting procedure. Cytotherapy 2006, 8, 166–177; Shah, F.S.; Li, J.; Zanata, F.; Curley, J.L.; Martin, E.C.; Wu, X.; Dietrich, M.; Devireddy, R.V.; Wade, J.W.; Gimble, J.M. The relative functionality of freshly isolated and cryopreserved Human Adipose-Derived Stromal/Stem Cells. Cells Tissue Organs 2016, 201, 436–444)

R: We appreciate the reviewer’s comment and suggestion of papers. In the revised manuscript, we deleted the contents of the parentheses that contain only fragmentary information and explained the conflicting issues in CD34 and STRO-1 expression on ASC with appropriate references at the end of the relevant paragraph.

Reviewer 2 Report

focus mainly on the interest of exosomes derived from ASCs in neurological disorders. But the potential interest of these products for other processes such as, for example, ischemic myocardium (Cui et al., J. Cardiovasc. Pharmacol. 2017), liver fibrosis (Qu et al., J. Cell. Mol. Med. 2017) or increased chemosensitivity in hepatocellular carcinoma (Lou et al., J. Hematol. Oncol. 2015) , should be mentioned. In any case, the authors could mention that a special interest of exosomes for the potential treatment of neurological disorders is their ability to cross the blood-brain barrier, which may be of interest to some readers.

- A major criticism of this section is that the authors almost limit themselves to describing known data about the therapeutic potential of exosomes, but not about the main objective of their review, which is how to improve their therapeutic potential, over which they go too far. .

-The authors emphasize the limitations of cell therapy and the potential of the products derived from its secretome: “…conventional cell-based therapy has several hurdles and limitations need to be considered for practical application”… Therefore, cell-free exosomes have attracted significant attentions as a novel therapeutic option in recent years”. However, they make no mention of the average conditioned interest and its soluble factors (For example, see: Vizoso et al., Int J Mol Sci. 2017).

Author Response

Reviewer #2

focus mainly on the interest of exosomes derived from ASCs in neurological disorders. But the potential interest of these products for other processes such as, for example, ischemic myocardium (Cui et al., J. Cardiovasc. Pharmacol. 2017), liver fibrosis (Qu et al., J. Cell. Mol. Med. 2017) or increased chemosensitivity in hepatocellular carcinoma (Lou et al., J. Hematol. Oncol. 2015) , should be mentioned. In any case, the authors could mention that a special interest of exosomes for the potential treatment of neurological disorders is their ability to cross the blood-brain barrier, which may be of interest to some readers.

R: We thank this reviewer for the helpful comments. In the revised manuscript, we added more citations reporting therapeutic potential of EV applications for other pathologic conditions as suggested by the reviewer.

- A major criticism of this section is that the authors almost limit themselves to describing known data about the therapeutic potential of exosomes, but not about the main objective of their review, which is how to improve their therapeutic potential, over which they go too far.

R: Since stem-cell derived EVs have been suggested as a novel and essential options in stem cell therapeutic field recently, we decided to add a subsection dedicated to ASC-EVs. During the revision, we clarified the fact that functional enhancement strategies for ASCs described in this review including genetic manipulation, modification of culture condition and priming with stimulants can also be applicable for producing the therapeutically superior EVs. Researchers have also been tried to improve EV functions via direct engineering and modification of EVs; however, we briefly mentioned about the future direction of EV application and added some of the comprehensive reviews regarding this topic instead not to blur this review`s focus.

-The authors emphasize the limitations of cell therapy and the potential of the products derived from its secretome: “…conventional cell based therapy has several hurdles and limitations need to beconsidered for practical application”… Therefore, cell-free exosomes have attracted significant attentions as a novel therapeutic option in recent years”. However, they make no mention of the average conditioned interest and its soluble factors (For example, see: Vizoso et al., Int J Mol Sci. 2017).

R: We understand the reviewer`s point. In the revised manuscript, we briefly summarized the major EV-derived beneficial factors and added some of the comprehensive reviews regarding this topic including suggested paper as follows;

In general, MSC-derived EVs (MSC-EVs) are isolated from MSC culture media via serial ultracentrifuge procedure then applied to the treatment for a wide range of abnormal pathologic conditions in which MSCs have proven to be effective. Recent advances in ‘omics’ technologies enable researchers to define the therapeutic candidates among the MSC-secreted paracrine factors. Since one of the most significant roles of EVs is to mediate the horizontal transfer of parent cell-derived signaling molecules to target cells, MSC-derived beneficial molecules such as TGF-β1, IL-10, PGE2, NO and IDO could be delivered via EVs. Therefore, the therapeutic actions of MSC-EVs are largely dependent on their tissue regenerative- and immunomodulatory capacity as MSCs. 

Reviewer 3 Report

This is an excellent review of the different procedures to control adipose stem cell functionalities. These methods could be fundamental tools to a variety of therapeutic applications, ranging from tissue regeneration to immune modulation in inflammatory diseases. It is a well written review, with the appropriate literature and I appreciate the tables that summarize several aspects of this heterogenic field, and thus help the reader to localize crucial information

I just have a couple of minor comments

1)      I would have liked a 3.2.3 section that perhaps addresses the genetic manipulation to induce lineage transdifferentiation by overexpression of transcriptional factors as for instance in the following works (see below). If there are similar approaches directed to other tissue regenerative approaches, it could be nice to summarize them (ie the transcriptional factors used, the lineage/s obtained, the level of functionality of the ASC-derived issue).

Neshati V, Mollazadeh S, Fazly Bazzaz BS, de Vries AA, Mojarrad M, Naderi-Meshkin H, Neshati Z, Kerachian MA. Cardiomyogenic differentiation of human adipose-derived mesenchymal stem cells transduced with Tbx20-encoding lentiviral vectors. J Cell Biochem. 2018 Jul;119 (7):6146-6153.

Goudenege S., Pisani D. F., Wdziekonski B., Di Santo J. P., Bagnis C., Dani C., et al. (2009). Enhancement of myogenic and muscle repair capacities of human adipose-derived stem cells with forced expression of MyoD. Mol. Ther. 17, 1064–1072. 10.1038/mt.2009.67

2)      In table 3, is the reference 107 correct for the hindlimb ischemia model?

Author Response

Reviewer #3

This is an excellent review of the different procedures to control adipose stem cell functionalities. These methods could be fundamental tools to a variety of therapeutic applications, ranging from tissue regeneration to immune modulation in inflammatory diseases. It is a well written review, with the appropriate literature and I appreciate the tables that summarize several aspects of this heterogenic field, and thus help the reader to localize crucial information

I just have a couple of minor comments

1)      I would have liked a 3.2.3 section that perhaps addresses the genetic manipulation to induce lineage transdifferentiation by overexpression of transcriptional factors as for instance in the following works (see below). If there are similar approaches directed to other tissue regenerative approaches, it could be nice to summarize them (ie the transcriptional factors used, the lineage/s obtained, the level of functionality of the ASC-derived issue).

Neshati V, Mollazadeh S, Fazly Bazzaz BS, de Vries AA, Mojarrad M, Naderi-Meshkin H, Neshati Z, Kerachian MA. Cardiomyogenic differentiation of human adipose-derived mesenchymal stem cells transduced with Tbx20-encoding lentiviral vectors. J Cell Biochem. 2018 Jul;119 (7):6146-6153.

Goudenege S., Pisani D. F., Wdziekonski B., Di Santo J. P., Bagnis C., Dani C., et al. (2009). Enhancement of myogenic and muscle repair capacities of human adipose-derived stem cells with forced expression of MyoD. Mol. Ther. 17, 1064–1072. 10.1038/mt.2009.67

R: We appreciate the reviewer’s helpful suggestion. As the reviewer suggested, the transdifferentiation/lineage conversion of MSCs through the TF gene transduction can be a breakthrough in terms of regenerative medicine. However, most studies have been conducted using BM-MSCs and do not have many cases of ASC, which is why we had tried to focus more on immunomodulatory and survival properties in our original manuscript. In the revised manuscript, accepting the reviewer’s opinion, we added a 3.2.3 section that briefly summarizes the case of ASC transdifferentiation to date as below;

3.2.3 Genetic manipulation to induce lineage transdifferentiation

In addition to the three mesenchymal lineages, ASCs can also undergo transdifferentiation toward non-mesenchymal cell lineages, including myogenic, cardiac, endothelial and neuronal, in response to the lineage-specific inducer [37,122]; although there are somewhat controversial views on neural transdifferentiation (reviewed in [123]). Lineage conversion can be achieved in vitro through the exposure of ASCs to extrinsic signaling molecules or via modification of culture conditions such as using specific biomaterials. Alternatively, genetic manipulation integrating key transcriptional factors into ASCs may be a better way to induce stable and effective lineage transdifferentiation [124]. Although there are still obstacles to be overcome such as the development of safe gene delivery methods and selection of the most appropriate target gene, encouraging evidence has been accumulated over MSCs from different sources. In this section, we will summarize the approaches to genetic manipulation to induce lineage transdifferentiation by overexpression of transcriptional factors in ASCs.

In case of cardiomyogenic lineage, forced expression of Tbx20, a critical transcription factor that contributes to heart development and cardiomyocyte regeneration, efficiently induced expression of cardiomyogenic differentiation markers on ASCs at 14 days after transduction both at the RNA and protein level [125]. It might be necessary to evaluate the cardiomyogenic regenerative capacity of Tbx20-overexpressed ASCs in ischemic heart disease animal model. To transdifferentiate ASCs toward the neural lineage, Tang et al., transduced the proneural transcription factor Neurogenin (Ngn2) into ASCs and evaluated in vitro neural lineage differentiation capacity and in vivo functional recovery in rat spinal cord injury (SCI). Rats transplanted with Ngn2-transduced ASCs showed higher expression of the neuron-specific nuclear protein (NeuN) in the injured site and exhibited the most striking functional recovery of the hind limb [126]. To enhance myogenic differentiation, Goudenege and colleagues transduced the key myogenic gene MyoD into human multipotent adipose-derived stem (hMADS) cells and observed a marked myogenic differentiation capability in vitro. Importantly, local intramuscular injection of MyoD-overexpressed hMADS cells into the cryoinjured Rag2-/-gC-/- immunosuppressed mice significantly improved muscle repair with the increase in hMADS-derived muscle fiber [127]. Moreover, it has been reported that ETS variant 2 (ETV2) overexpression in ASC can generate functional and expandable ETV2-induced endothelial-like cells (EiECs), which is expected to be an alternative strategy to treat ischemic vascular disorders [128].

2)      In table 3, is the reference 107 correct for the hindlimb ischemia model?

R: We thank this reviewer for pointing out our adventitious errors. Including this point, we evaluated manuscript carefully and corrected all of citation number errors.

Reviewer 4 Report

The current manuscript addresses the strategies to enhance the functionality of adipose stem cells. In my opinion it’s a very important topic as future MSC therapeutics should include highly viable and functional MSC populations for better outcome. The present manuscript is scientifically accurate; authors have described adequately the previous literature concerning techniques/strategies to improve MSC functionality, viability and migration, and have presented selected up to date references to cover all these issues. In my opinion the present manuscript is of importance for the reader.

Please find below a number of comments that authors should take into consideration:

Abstract, Paragraph 1, line 13: Instead of using the term ‘reagent’ maybe authors can use another term such as ‘therapeutic tool’

Lines 48-49: Authors should provide some references regarding the heterogeneity of MSC populations.

Line 88: Authors should add the ‘low’ MSC percentage they are referring to so the reader will have an exact idea of the numbers.

Lines 100-103: Authors can add few sentences regarding the tissue types and techniques used to isolate ASCs.

Line 102: Authors should add the ASC percentage they are referring to so the reader will have an exact idea of the numbers.

Line 109: Authors stated that STRO-1 expression is totally negative in ASCs. Authors should add more recent published work (Zannettino et al., 2008) that show somw expression of STRO-1 in ASCs. Also authors could add the pioneering thorough study of Zimmerlin et al., 2013 that describes the different ASC subpopulations.

Lines 128-129: This statement that ‘with no exception’ ASCs failed to have positive outcome in vivo is a very strong statement that is not totally true. There are studies using ASCs with very important clinical outcome; for example Fang studies for GvHD. I think authors should be more reluctant in this sentence.

Line 229: Authors should define which TOLL-like receptors are expressed from MSCs and how their expression is associated with tissue of origin.

Line 243: Authors should comment if TLR priming except immunosuppression is affecting MSC differentiation capacity (such as Lombardo et al. work).  

Lines 306-314: This paragraph can be easily transferred to the ‘hypoxia’ separate section.

Line 361: Authors should remove the typographic error ‘ad’.

Table 3: Authors should double check the reference numbering as some referenced articles within the table are review articles. Are all the studies mentioned in the table about ASCs or the table refers to MSCs in general?

Line 508: This section is well written however authors should add a table to clarify and describe better the studies that are using EVs and the type of model they are applying EVs to (similar table to 3D spheroids table).  

Author Response

Reviewer #4

The current manuscript addresses the strategies to enhance the functionality of adipose stem cells. In my opinion it’s a very important topic as future MSC therapeutics should include highly viable and functional MSC populations for better outcome. The present manuscript is scientifically accurate; authors have described adequately the previous literature concerning techniques/strategies to improve MSC functionality, viability and migration, and have presented selected up to date references to cover all these issues. In my opinion the present manuscript is of importance for the reader.

Please find below a number of comments that authors should take into consideration:

Abstract, Paragraph 1, line 13: Instead of using the term ‘reagent’ maybe authors can use another term such as ‘therapeutic tool’

R: As suggested by this reviewer, we modified expression in the revised manuscript.

Lines 48-49: Authors should provide some references regarding the heterogeneity of MSC populations.

R: Based on the reviewer’s comment, we added two comprehensive reviews as references for the MSC heterogeneity along with one sentence for the smoother flow as follows;

The heterogeneity of MSCs can be attributed to multiple factors including among donors, tissue origins and even subpopulation within the same origin (reviewed in [19, 20]).

Line 88: Authors should add the ‘low’ MSC percentage they are referring to so the reader will have an exact idea of the numbers.

R: As suggested by this reviewer, we indicated the range (0.001% - 0.1% in mononuclear fraction) of low BM-MSCs yield in the corresponding sentence along with the relevant reference.

Lines 100-103: Authors can add few sentences regarding the tissue types and techniques used to isolate ASCs.

R: We appreciate the reviewer’s comment and agree that this paragraph is a bit ambiguous to apparently describe the adipose tissues, SVF and ASC. In the revised manuscript, we added a few sentences explaining the adipose tissue type, isolation technique, and characteristics of ASC in order to clarify the reader’s understanding.

Line 102: Authors should add the ASC percentage they are referring to so the reader will have an exact idea of the numbers.

R: We added relative fold change in MSC yield between adipose tissue and bone marrow in line 103 of the revised manuscript as follows;

Moreover, adipose tissue can provide about 500-fold higher number of MSCs than from an equivalent amount of BM aspirates [32].

Line 109: Authors stated that STRO-1 expression is totally negative in ASCs. Authors should add more recent published work (Zannettino et al., 2008) that show somw expression of STRO-1 in ASCs. Also authors could add the pioneering thorough study of Zimmerlin et al., 2013 that describes the different ASC subpopulations.

R: We appreciate the reviewer’s comment and suggestion of papers. In the revised manuscript, we deleted the contents of the parentheses that contain only fragmentary information and explained the conflicting issues in CD34 and STRO-1 expression on ASC with appropriate references at the end of the relevant paragraph as follows;

Adipose tissue is an extremely heterogeneous tissue composed of various cell types, including ASCs, preadipocyte, adipocyte, fibroblast, vascular smooth muscle cells, endothelial cells and lymphocytes [33]. Multipotent stem cells can be obtained from subcutaneous or visceral fat tissue harvested through liposuction or surgical resection procedure and are known to reside mainly in the perivascular region of white adipose tissue [34]. Indeed, CD146+ CD34- pericytes and CD146-CD34+ supra-adventitial adipose stromal cells have been revealed as two distinct MSC subsets in human adipose tissue with similar characteristics to BM-MSC [35]. The term ASC refers to all kinds of adipose-derived stem cells and ASCs are mainly contained in a stromal vascular fraction (SVF) which are yielded by mechanical and enzymatic digestion [25,36,37]. This heterogeneous SVF could be applied directly to the regenerative medicine (reviewed in [38]) or further cultured to isolate and expand the pure adherent ASC fraction [39]. Given the fact that ASCs primarily reside around vasculature in fat tissue, scientists have sought to verify whether what we refer to as ASC is the inherent MSC originated from adipose tissue or unknown cells from a mesodermal origin that migrates from peripheral blood to adipose tissue [40]. A series of comparative studies have conducted in MSCs from different origin and revealed the evidence that ASCs are unique MSCs of adipose tissue based on a difference in the degree of differentiation potential and phenotypic marker expression [41,42]. However, there is still controversy over the expression of specific markers in ASCs; some studies have reported positive CD34 expression on ASCs [43,45], whereas a few studies have shown either absent or disappear by passaging or cryopreservation [46]. STRO-1, which is considered one of the stemness markers of MSCs, was found to be negative in ASCs, distinct from BM-MSCs [44], but more recently, there has been a conflicting report that ASC subsets express STRO-1 [47]. Further studies must be warranted to identify the precise stem cell subpopulation within adipose tissue in terms of the phenotype and distinct biologic functions.

Lines 128-129: This statement that ‘with no exception’ ASCs failed to have positive outcome in vivo is a very strong statement that is not totally true. There are studies using ASCs with very important clinical outcome; for example Fang studies for GvHD. I think authors should be more reluctant in this sentence.

R: We agree with the reviewer’s opinion and it’s also far from what we’re trying to say. In the revised manuscript, we rephrased this sentence more smoothly as below;

Despite these favorable pre-clinical data, however, ASCs have shown somewhat limited outcome and fallen short of the expectations in advanced clinical trials (highlighted in [60]).

Line 229: Authors should define which TOLL-like receptors are expressed from MSCs and how their expression is associated with tissue of origin.

R: Based on the reviewer’s suggestion, we added both detailed explanations and linked references regarding what type of TLRs are expressed in MSCs and what differences are there between tissue origins.

Line 243: Authors should comment if TLR priming except immunosuppression is affecting MSC differentiation capacity (such as Lombardo et al. work).  

R: As the reviewer pointed out, we added a paragraph describing the impact of TLR priming on MSC differentiation potential before the beginning of the immunomodulation-related paragraph.

Ligation of TLR with specific agonist can serve as modulators of ASC multi-lineage differentiation capacity. Cho et al., [84] showed that activation of TLR2 (by peptidoglycan, PGN) and TLR4 (by lipopolysaccharide, LPS) significantly enhanced the osteogenic differentiation in a dose-dependent manner, whereas triggering TLR9 (by CpG oligodeoxynucleotide, CpG-ODN) inhibited osteogenesis and ASC proliferation. Activation of TLR3 (by polyinosinic-polycytidylic acid, poly I:C) and TLR5 (by flagellin) did not cause any changes. Adipogenesis of ASCs was remarkably inhibited only in the presence of PGN. Similarly, pre-conditioning ASCs with poly I:C and LPS enhanced osteogenic differentiation without any effects on adipogenic differentiation and self-renewal [89]. The degree of altered differentiation potential may be different depending on the MSC source, even if the same TLR is stimulated [90], and the impact on other lineages such as chondrogenic differentiation has not been clarified yet. Thus, more in-depth comparative investigations would need to solve this question.

Lines 306-314: This paragraph can be easily transferred to the ‘hypoxia’ separate section.

R: We appreciate the reviewer’s comment. To prevent duplicate the same contents, this paragraph was moved to at the end of the introductory paragraph in 3.3.2. Hypoxic treatment section (from line 535 in the revised manuscript).

Line 361: Authors should remove the typographic error ‘ad’.

R: We appreciate the reviewer’s detailed proofreading and comment. As the reviewer pointed out, we deleted the typographic error.

Table 3: Authors should double check the reference numbering as some referenced articles within the table are review articles. Are all the studies mentioned in the table about ASCs or the table refers to MSCs in general?

R: We thank this reviewer for pointing out our adventitious errors. Including this point, we evaluated manuscript carefully and corrected all of citation number errors. Also, all of the studies summarized in this table are about ASC application.   

Line 508: This section is well written however authors should add a table to clarify and describe better the studies that are using EVs and the type of model they are applying EVs to (similar table to 3D spheroids table).  

R: We agreed with this reviewer`s opinion and created new table summarizing the representative works proving the therapeutic benefits of ASC-EVs. 

Round 2

Reviewer 2 Report

Part of the review report was not answered. I do not know if the report was not forwarded completely to the authors.

Abstract:

-Line 18: The term “compatible” does not seem adequate, it should be “comparable”, for example.

1.Introduction:

-Line 53: The expression “…in the early 2000s” is the expression is inaccurate, it should be replaced by 2006.

-Lines 62-63: The authors should provide data on how much the cells survive in the cultures once transplanted, with the corresponding references. It has been reported that that <1% MSCs survive for more than one week after systemic administration (Lee et al., 2009;5:54–63; Parekkadan et al.; Eggenhofer et al. Front. Immunol. 2012;3:297; Song et al., Cell Transplant. 2012;21:1641–1650).

-The authors had to better justify the need for strategies to improve the role of ASCs.

2.Adipose stem cells

2.1.Characteristiss of ASC

-Almost the entire first paragraph is referred to the MSCs in general. That information should be transferred to the introduction section.

2.2.ASCapplications and limitations

-In this sense, the authors should mention the problem of the heterogeneity and limitations of the ASCs, in relation to the types, anatomical location, mode of obtaining, and conditions of the donors, such as age, obesity, diabetes, etc.

3.1.3. Other bioactive molecules and combination therapy

-These sections are very extensive and repetitive. They should be considerably shortened and better structured, eliminating information about conflicting or irrelevant results.

3.2. Genetic manipulation

-These sections are very extensive and speculative. The authors should significantly shorten the text, merely sticking to the data the objective data.

3.3.Modification of cell culture conditions

-Certain aspects related to this section are omitted, such as the influence of other biomaterial scaffolds, rigidity or stiffness of the surface, as well as medium supplements (For example see: Prieto González EA. Heterogeneity in Adipose Stem Cells. Adv Exp Med Biol.2019;1123:119-150).

Author Response

Comments and Suggestions for Authors

Part of the review report was not answered. I do not know if the report was not forwarded completely to the authors.

R: Actually, we did not receive some of this reviewer`s comments in the first round of revision and we feel sorry for the missing part. In this revision, we tried our best to cover the reviewer`s suggestions as follows.

Abstract:

-Line 18: The term “compatible” does not seem adequate, it should be “comparable”, for example.

R: As suggested by the reviewer, we changed the expression to ‘comparable’ in the revised abstract.

1.Introduction:

R: We appreciate the reviewer’s insightful comments. As we agree with the reviewer’s suggestion, both 1. Introduction and 2. Adipose stem cell sections were almost fully re-written in this revision by deleting unnecessary parts and blending repetitive information only into the Introduction section. Point-by-point responses to each following comment will be answered based on this revision.

-Line 53: The expression “…in the early 2000s” is the expression is inaccurate, it should be replaced by 2006.

R: In the process of revising the section 1 and 2, we decided to remove the 2006 ISCT criteria for MSC standardization and instead referred to the 2013 IFATS and ISCT publication specified to ASC population in order to clarify what we want to say in this manuscript. This correction corresponds to the lines 64-68.

-Lines 62-63: The authors should provide data on how much the cells survive in the cultures once transplanted, with the corresponding references. It has been reported that that <1% MSCs survive for more than one week after systemic administration (Lee et al., 2009;5:54–63; Parekkadan et al.; Eggenhofer et al. Front. Immunol. 2012;3:297; Song et al., Cell Transplant. 2012;21:1641–1650).

R: We thank for the helpful comment. We totally agree that low distribution and short survival period after systemic administration are important issues in MSC therapy. As suggested by the reviewer, this content was described from lines 58 and the corresponding references were added.

-The authors had to better justify the need for strategies to improve the role of ASCs.

R: As the reviewer pointed out, we reorganized the Introduction part and added a sentence in the last paragraph in order for readers to more clearly understand the need for strategies to enhance ASC functionality.

2.Adipose stem cells

2.1.Characteristiss of ASC

-Almost the entire first paragraph is referred to the MSCs in general. That information should be transferred to the introduction section.

R: This part was blended into the Introduction section as mentioned above and replaced by the description of ASC’s overall characteristics including the terminology, tissue source types, isolation methods, variability among donors with obese, aged and diabetes.

2.2.ASCapplications and limitations

-In this sense, the authors should mention the problem of the heterogeneity and limitations of the ASCs, in relation to the types, anatomical location, mode of obtaining, and conditions of the donors, such as age, obesity, diabetes, etc.

R: We appreciate the reviewer’s comment. We explained this content but added it to the section of 2.1. Characteristics of ASCs. In the part 2.2. ASC applications and limitations, we filled with the practical ASC applications and their limitations, instead.

3.1.3. Other bioactive molecules and combination therapy

-These sections are very extensive and repetitive. They should be considerably shortened and better structured, eliminating information about conflicting or irrelevant results.

R: As suggested by the reviewer, we changed the title of this section to “Pre-conditioning with other pharmacological/bioactive molecules” and deleted unnecessary and ambiguous part such as combination therapy ASC with drug.

3.2. Genetic manipulation

-These sections are very extensive and speculative. The authors should significantly shorten the text, merely sticking to the data the objective data.

R: As suggested by the reviewer, we shortened the entire 3.2 section and tried to present this section objectively based on the clear data. According to the other reviewer’s suggestion in round 1, we created ‘3.2.3. Genetic manipulation to induce lineage transdifferentiation’ subsection, which addresses the gene modification strategies to induce lineage transdifferentiation into cardiac, neural, myogenic and endothelial lineages.

3.3.Modification of cell culture conditions

Certain aspects related to this section are omitted, such as the influence of other biomaterial scaffolds, rigidity or stiffness of the surface, as well as medium supplements (For example see: Prieto González EA. Heterogeneity in Adipose Stem Cells. Adv Exp Med Biol.2019;1123:119-150).

R: As suggested by the reviewer, we created a new subsection titled ‘3.3.3. Other culture environmental modifications’, which briefly summarizes other approaches to manipulate the culture conditions including media formula changes and modifying the stiffness of culture matrices.